# Chloramphenicol Resurrected: A Journey from Antibiotic Resistance in Eye Infections to Biofilm and Ocular Microbiota

**DOI:** 10.3390/microorganisms7090278

**Published:** 2019-08-21

**Authors:** Lorenzo Drago

**Affiliations:** Clinical Microbiology, Department of Biomedical Science for Health, University of Milan, 20133 Milan, Italy; lorenzo.drago@unimi.it

**Keywords:** chloramphenicol, antibiotic resistance, ocular infections, eye bacterial biofilms, ocular microbiota

## Abstract

The advent of multidrug resistance among pathogenic bacteria is devastating the worth of antibiotics and changing the way of their administration, as well as the approach to use new or old drugs. The crisis of antimicrobial resistance is also due to the unavailability of newer drugs, attributable to exigent regulatory requirements and reduced financial inducements. The emerging resistance to antibiotics worldwide has led to renewed interest in old drugs that have fallen into disuse because of toxic side effects. Thus, comprehensive efforts are needed to minimize the pace of resistance by studying emergent microorganisms and optimize the use of old antimicrobial agents able to maintain their profile of susceptibility. Chloramphenicol is experiencing its renaissance because it is widely used in the treatment and prevention of superficial eye infections due to its broad spectrum of activity and other useful antimicrobial peculiarities, such as the antibiofilm properties. Concerns have been raised in the past for the risk of aplastic anemia when chloramphenicol is given intravenously. Chloramphenicol seems suitable to be used as topical eye formulation for the limited rate of resistance compared to fluoroquinolones, for its scarce induction of bacterial resistance and antibiofilm activity, and for the hypothetical low impact on ocular microbiota disturbance. Further in-vitro and in vivo studies on pharmacodynamics properties of ocular formulation of chloramphenicol, as well as its real impact against biofilm and the ocular microbiota, need to be better addressed in the near future.

## 1. Introduction

Antibiotic resistance is a worldwide concern because of its current and possible future impact on global health and the costs of national healthcare systems, mainly due to more limited treatment options [1]. It has been suggested that the number of infections caused by resistant microbes is increasing [2], and that there is a need for a precise estimate of the burden of antibiotic resistance in developed countries in order to be able to introduce interventional strategies capable of limiting or preventing the spread of resistant infections [3].

The bacterial strains resistant to the majority of antibiotics (commonly known as “super-bugs”) are not only involved in cases of pneumonia and infections affecting the urinary tract and skin, but also in often clinically neglected eye infections. The main culprits are *Staphylococcus aureus* (and more recently, *Staphylococcus epidermidis*), which are known to be resistant to methicillin, and *Neisseria gonorrhoeae*, which show increased resistance to penicillin, tetracycline, and, particularly, ciprofloxacin. However, other frequently encountered bacteria, such as *Enterococcus faecalis*, *Acinetobacter baumannii*, or *Pseudomonas aeruginosa*, have become increasingly resistant to some relatively new antibiotics as the latest-generation cephalosporins, carbapenems against Gram-negative bacteria, and daptomicin and glycopetides against Gram-positive bacteria. Similarly, this happened to the older antibiotics that are now sometimes seen as “last-chance drugs”, such as colistin, that can still be used against Gram-negative bacteria despite the clinical appearance of bacteria harboring the colistin-resistance mcr-1 gene [4]. There has also been a dramatic increase in the resistance of *Enterobacteriaceae* (especially *Escherichia coli* and *Klebsiella* spp.), and recent data have indicated an increase in bacterial enzymes that give rise to multi-resistance to bectalactams, such as extended-spectrum β-lactamases (ESBL), and carbapenems, such as carbapenemases (KPC) [5,6].

As if these factors complicating infection management were not enough, there is the additional problem of bacterial biofilms: i.e., well-structured consortia of bacteria embedded in a self-produced polymeric matrix consisting of polysaccharide, protein, and genetic material. Biofilm-related infections are generally chronic because they show increased tolerance to many antibiotics: for example, the persistence of staphylococcal or pseudomonal infections is often due to biofilm formation [7].

Antibiotic resistance and biofilms are also associated with the pathogens causing eye infections, and factors such as prescribing antibiotics without identifying the germ, short-term treatments, and repeated exposure to the same antibiotic are now considered the main contributors to antibiotic resistance and treatment failure [8].

Next-generation sequencing (NGS) has also recently evidenced a well-balanced ocular microbiota (the resident ocular microflora including a network of several species) protecting the eyes surface, and that, the changes in the resident eye microbiota induced by antibiotic treatment seem to unbalance the entire ocular system and promote the onset of infection [9].

Chloramphenicol is a broad-spectrum antibiotic that had been partially abandoned in developed countries because its systemic administration is associated with fatal aplastic anemia [10] but is now widely used in response to the continuing problem of multi-drug resistant pathogens [11]. Chloramphenicol has been indeed often blamed for being a possible cause of aplastic anemia by postulating that the p-NO2 group of chloramphenicol is the structural feature underlying this adverse event [12], especially in case of viral hepatitis [13,14]. This last eventuality causes an abnormal metabolism of this molecule, and the reduction of the nitrobenzene ring to a nitroso group on chloramphenicol may lead to potential DNA damages in the stem cell, and potentially to cell death. Other studies have hypothesized that, also, particular bacteria of the gut microbiota may metabolize chloramphenicol to toxic metabolites [15].

Chloramphenicol and its fluorinated derivative florfenicol represent highly potent inhibitors of bacterial protein biosynthesis. As a consequence of the use of this drug in human and veterinary medicine, bacterial pathogens of various species and genera have developed and/or acquired resistance. Several mechanisms responsible for resistance to chloramphenicol can occur, i.e., pump efflux, acetyltransferases, or transposons, and other mobile genetic elements carrying resistance genes [16].

However, although the systemic role of chloramphenicol has been re-assessed [17], it has recently been found that its in-vitro anti-fungal activity is comparable with that of other anti-fungal compounds [18], and that the non-toxic topical formulation requires further comparisons with other topical drugs in the light of the new antibiotic resistance of ocular strains.

The aim of this review is to analyze the role of ocular topical chloramphenicol in the current context of increasing antimicrobial resistance rates and the emergence of ocular biofilm-related infections and consider the implications of these findings for clinical ophthalmology. The antibiofilm findings of chloramphenicol, as well as the complexity of bacterial ocular communities, e.g., the microbiota, and impact on this ecosystem by the antimicrobial drugs will also be briefly discussed.

## 2. Microorganisms Responsible of Eye Infections

Bacterial eye infections include endophthalmitis, conjunctivitis, keratitis, blepharitis, orbital cellulitis, and dacryocystitis. Endophthalmitis is often caused by adnexal microbial flora after cataract surgery or repeated ocular injections [19] and, although rare, can have extremely serious consequences. It is therefore essential to discover the pathogen involved, its antibiotic susceptibility, and the local and regional distribution of antibiotic resistance in order to ensure effective treatment. However, although the treatment guidelines for eye infections recommends a laboratory culture before drug administration, many clinicians start antibiotic therapy without any etiological information [20]. This means that little is known about the epidemiology of causative pathogens other than what has been learned from the few clinical studies carried out in some problem-sensitive countries.

Although more attention is now being given to microorganisms having a propensity to induce resistance in the community because of their incidence and/or the incorrect use of antibiotics (i.e., *S. aureus*, *P. aeruginosa* and *S. epidermidis*) [21], it is important to remember that the microorganisms responsible for eye infections may change with the site of infection and involve both Gram-positive and Gram-negative bacteria. Endophthalmitis is generally caused by coagulase-negative *Staphylococcus* (CoNS), but may also be caused by *Streptococcus* spp and *Bacillus* spp, especially in the case of trauma and injury [22,23], whereas the most frequently isolated pathogens in cases of bacterial conjunctivitis are *S. aureus*, *S. pneumoniae*, *Haemophilus influenzae*, and, less frequently, *S. epidermidis*, *Enterococcus* spp., the Streptococci viridans group, *E.coli*, *Serratia marcescens*, *P. aeruginosa*, and *Proteus mirabilis* [24]. Similar epidemiology is observed in patients with keratitis, which may be caused by Gram-positive *S. aureus*, *S. epidermidis*, and several Streptococci, or Gram-negative *P. aeruginosa*, *S. marcescens*, *Moraxella* spp, and *H. influenzae* [25]. The most frequent causes of blepharitis are *S. aureus* and CoNS [26], whereas the typical localization of orbital cellulites means that it is largely due to *S. aureus*, *Streptococcus pyogenes*, and *H. influenzae* [27]. Dacryocystitis, which is due to obstruction of the naso-lacrimal duct, is caused by some dangerous Gram-negative bacteria such as *P. aeruginosa*, *E. coli*, *Enterobacter aerogenes*, and *Citrobacter* spp, as well as by Gram-positive bacteria such as *S. aureus*, *S. pneumoniae*, and *Enterococcus* spp [28].

More recent data have indicated that *S. aureus* and CoNS (especially *S. epidermidis*) are the main etiological agents of eye infections, but their incidence is different in keratitis (*S. aureus* >25%) and endophthalmitis (*S. epidermidis* >30%). Finally, *S. pneumoniae* and *H. influenzae* are frequently isolated from patients with conjunctivitis, and *S. aureus* and *P. aeruginosa* are a leading cause of contact lens-related bacterial keratitis [29,30,31,32,33].

This particular bacterial epidemiology imposes the use of a broad-spectrum antibiotic in the empirical treatment of eye infections. Pharmacodynamically, topical antibiotics are more effective in rapidly ensuring high concentrations at the site of infection, and may therefore be more likely to inactivate the microorganism at an early stage of infection and during the chronic phase, although their prolonged use may increase the overall burden of antibiotic resistance [34].

## 3. Burden of Antibiotic Resistance to Ocular Drugs

Some of the main goals of the American centers for disease control and prevention and the world health organization include promoting the judicious use of antibiotics and limiting the increase in antimicrobial resistance, and so they have proposed national and international surveillance programs and adequate training in order to improve awareness of antibiotic susceptibility patterns. In terms of eye infections, the ocular tracking resistance in the US today (TRUST) program merits particular mention, as it annually evaluates the in-vitro antibiotic susceptibility of *S. aureus*, *S. pneumoniae*, and *H. influenzae.* This surveillance system has shown that ciprofloxacin resistance has increased over the last ten years [35], and that the trend of the methicillin resistance (MR) of *S. aureus* (MRSA) and *S. epidermidis* (MRSE) has been similar [36].

In this scenario, further consideration needs to be given to the use of antimicrobial prophylaxis for the prevention of post-operative infections in patients undergoing eye surgery. Although it is judged to be necessary in order to decrease the risk of infections such as endophthalmitis, some antibiotics may lead to changes in resident ocular flora and contribute to increasing the resistance of pathogens, such as Staphylococci and Gram-negative bacteria [18].

In 2004, a study by Marangon et al. [37] reported a worrying increase in the fluoroquinolone resistance of MRSA isolated from patients with conjunctivitis or keratitis. The rate of MSSA resistance to levofloxacin was 4.7% and that of resistance to ciprofloxacin was 11.9%, whereas the rate of resistance to the two fluoroquinolones in MRSA was 95.7% and 82.1%, respectively. As pointed out by Galvis et al., the different fluoroquinolones may have different patterns of resistance, and their use over time has dramatically increased the resistance of MR Staphylococci [38].

A study of a large number of *S. aureus*, CoNS, *Streptococcus* spp, and *Pseudomonas* spp isolated from patients with conjunctival and corneal infections [39] concluded that their susceptibility to gentamicin, tobramycin, and cephalothin had decreased, whereas the fluoroquinolones and chloramphenicol remain a good choice, although susceptibility to them varies depending on the microorganisms. Unfortunately, this study does not mention the rate of methicillin resistance among the isolated Staphylococci.

Shanmuganathan et al. determined the prevalence and clinical characteristics of external ocular infections caused by MRSA in an ophthalmic hospital in the UK and found that all MRSA were sensitive to chloramphenicol but not to ofloxacin in patients aged >50 years [40].

A recent antibiotic resistance monitoring in ocular microorganisms (ARMOR) report concerning the trends of the antibiotic resistance of four species responsible for eye infections in the US between 2009 and 2016 [41] showed resistance to azithromycin, ciprofloxacin, and methicillin in 60.6%, 35.8%, and 36.6% of the *S. aureus* isolates, respectively. Resistance to tobramycin and clindamycin was found in 17.4% and 15.4%, respectively, but, interestingly, only a few of the isolates were resistant to chloramphenicol (6.1%), trimethoprim (4.4%), or tetracycline (4.3%), and none of them were resistant to vancomycin. The MRSA strains also showed high levels of resistance to azithromycin, fluoroquinolones, tobramycin, and clindamycin. Similar results were observed in the case of CoNS, although, in comparison with *S. aureus*, a greater percentage were MR (48.6%) and a smaller percentage were resistant to chloramphenicol (1.2%). As expected, *S. pneumoniae* strains had high levels of in-vitro resistance to azithromycin (35.9%) and penicillin (33.3%), but not to the other tested drugs. *P. aeruginosa* showed a trend towards moderate cumulative resistance to polymyxin B (8.6%), tobramycin (2.5%), and ciprofloxacin (6.0%).

A Japanese group retrospectively analyzed the trend of resistance of *S. aureus*, CoNS, and *Corynebacterium* isolated from patients with eye infections [42] and observed a high prevalence of Staphylococci and Corynebacteria in those with conjunctivitis, keratitis, and dacryocystitis (80% in total). Resistance profiles revealed a generalized resistance of Staphylococci to methicillin that slightly varied over the years, and resistance to the fluoroquinolones that remained stable throughout the 10 years of observation.

Increased rates of resistance to the fluoroquinolones have also been reported by other authors [43]. Chang et al. [32] found an increase in the rate of resistance to the fluoroquinolones, especially in the case of MRSA, and it has recently been highlighted that MRSA has not only acquired resistance to vancomycin, but also that the resistance rate is higher in certain countries [44,45]. These and other findings suggest that caution is required when using fluoroquinolone eye drops to treat *Corynebacterium* keratitis in order to reduce the risk and prevalence of fluoroquinolone-resistant Corynebacteria [46].

## 4. Role of Biofilm in Eye Infections

The presence of biofilms has been reported in association with many eye infections, especially in people who wear contact lenses or those who have undergone cataract surgery with the placement of an intra-ocular lens or the introduction of intraocular infusion pumps, glaucoma tubes, stents, keratoplasties, or other ocular prostheses. These abiotic surfaces may create a favorable environment for the development of biofilm-related infections [47].

A biofilm is a polysaccharide-coated matrix that is self-produced by bacteria or fungi, which aggregate to form a microbial colony that attaches to a surface by means of a slimy layer that helps to protect the microorganisms themselves. There are a number of reasons for the formation of biofilms, all of which promote microbial growth and survival, and provide protection against almost any biological environment, particularly under negative conditions such as during antibiotic treatment. Biofilms adhere to surfaces that offer an additional source of nutrients and thus keep the microorganisms in optimal condition. Bacterial cells lie closely together, and cell–cell communications are facilitated by means of signaling molecules in a process known as quorum sensing.

A previously unknown clinical presentation of a corneal biofilm, consisting of superficial and recurrent corneal plaques, has recently been reported, which allows the prolonged survival of microorganisms even in the absence of prosthetic material, clinical signs, or symptoms of active corneal inflammation and/or infection [48]. It seems that bacteria can persist for long periods inside this structure, and thus create probably low-grade inflammation similar to that which occurs in the case of a typical biofilm-related infection [49].

The literature includes a number of papers describing keratitis and contact lens biofilm colonization, as well as biofilm-related conjunctivitis [50,51,52,53], all of which have the common characteristic of being difficult to treat and eradicate. Biofilms can be difficult to remove and can jeopardise human health by preventing the penetration of the immune system or antibiotics [54]. When bacteria are embedded in biofilm, they are 100–1000 times more tolerant to the antimicrobial agents than their corresponding planktonic cells, and one of the main limitations of antibiotics, such as ampicillin and ciprofloxacin, is their poor ability to reach the biofilm-embedded bacteria [55].

There is therefore increasing interest in improving the pharmacokinetic/pharmacodynamic (PK/PD) ratio of antibiotics in order to reduce the tolerance of biofilm-embedded bacteria. Topical administration can lead to high local concentrations by delivering antibiotics directly to the site of infection without giving rise to the serum concentrations that induce systemic side effects [56]. Hume et al. found that *S. marcescens* bacteria endowed with biofilm and colonizing a contact lens could resist phagocytosis [57], and it is not surprising that they are also more resistant to antibiotics. Ray et al. [58] investigated the susceptibility of *S. marcescens* biofilm to high doses of common antibiotics (ceftriaxone, kanamycin, gentamicin, and chloramphenicol) and other non-antimicrobial agents, and concluded that only chloramphenicol reduced biofilm biomass and viability. Furthermore, another study [59] has demonstrated that chloramphenicol can act synergistically with the antimicrobial peptides (AMPs), such as FK-13-a1 and FK-137, to reduce the bacterial biofilm produced by multidrug-resistant *P. aeruginosa*, vancomycin-resistant *E. faecalis*, and MRSA.

An antibiotic has to penetrate biofilm in order to be able to kill bacteria and avoid the development of resistant micro-colonies. Singh et al. [60] have evaluated the ability of various antibiotics to penetrate the biofilms produced by *S. aureus*, *S. epidermidis*, *E. coli*, and *K. pneumoniae*. They found that this was only achieved by vancomycin and chloramphenicol, and that this not only depended on the bacterial genus and strain, but also on the antibiotic. Similar results have been obtained by Liagat et al. [61], who have demonstrated that tetracycline and chloramphenicol are effective in reducing the biofilm formed by *Klebsiella* spp, *P. aeruginosa*, *Achromobacter* spp, *K. pneumoniae*, and *Bacillus pumilis*.

One crucial step in biofilm development is the initial interaction between bacteria and the abiotic or biotic surfaces that can ultimately lead to colonization and biofilm-related infections. Consequently, reducing adhesion is the strategy of choice for preventing biofilm formation [62], particularly in the case of contact lens contamination and consequent ocular keratitis [63]. Drago et al. [64] have studied the anti-microbial activity of tobramycin and chloramphenicol against *S. aureus*, *S. epidermidis*, *E. faecalis*, group A, B, and G Streptococci, *Klebsiella* spp, *Stenotrophomonas maltophilia*, and ciprofloxacin-resistant and ciprofloxacin-susceptible *P. aeruginosa*, and evaluated their ability to interfere with the adhesion of slime-producing strains of *S. aureus* and *P. aeruginosa* to intra-ocular lenses. The results showed that chloramphenicol was more active than tobramycin against Gram-positive bacteria, and that treating lenses with them prevented the formation of bacterial biofilms. Chloramphenicol, in comparison with tobramycin, was more active against *S. aureus*, and less active against *P. aeruginosa*, although chloramphenicol also significantly reduced bacterial adhesion even when the lenses were colonized by *P. aeruginosa*. The authors suggested that chloramphenicol could be used for topical prophylaxis in order to avoid bacterial contamination but stressed the need for further specific studies.

## 5. Microbiota: An “Organ” to Safeguard During Antibiotic Treatment

It has recently been postulated that there may be a link between the gut and the eyes [65]; for example, it is thought that the pathogenesis of inflammatory bowel disease (IBD) involves the microbiota of the gut (intestinal bacteria, viruses, fungi, and protozoans), but this microbiota can also cause distal clinical manifestations, such as episcleritis, uveitis, and conjunctivitis [66]. Leger [65] also describes the role of intestinal microbiota in triggering autoimmune retinal disease in a uveitis mouse model, and in triggering age-related macular degeneration, glaucoma, and diabetic retinopathy.

The existence of an eye microbiota is still debated. It is likely that the cornea does not contain a measurable number of microbes because it is a microbe-hostile environment, but other ocular sites are colonized by a typical microbiota that is capable of protecting the entire ocular system [67,68]. Prevalent commensal microorganisms include *Corynebacterium*, Staphylococci, and *Propionibacterium* in the conjunctiva, and *Pseudomonas*, *Acinetobacter*, and *Methylobacterium* in the eyelids. These surely protect their host against eye infections: for example, Leger et al. [69] have recently demonstrated an ocular immunological response to *Corynebacterium* mastitis by showing that the commensal has specific activity against corneal infections caused by *P. aeruginosa* and *C. albicans* by inducing an interleukin 17 response from mucosal γδ T cells.

Eye infections can be caused by endogenous (resident) or exogenous (environmental) microbes, which raises the questions as to whether commensal microorganisms can counteract the pathobionts that cause opportunistic infections, and whether topical antibiotics disrupt this finely protective ocular equilibrium. One elegant animal experiment has been conducted by Kugadas et al. [70], who demonstrated that a β-lactam antibiotic such as cefazolin can induce dysbiosis in the ocular microbiota of house finches; it also caused more severe infection-induced tissue damage and enhanced the pathogenic virulence of *Mycoplasma gallisepticum*. The authors concluded that antibiotic perturbation increased the extent of conjunctival inflammation during infection. Combined with recent evidence that the ocular microbiota protects against *P. aeruginosa*-induced keratitis in mice, these findings suggest that the ocular microbiota has protective functions similar to that of a number of other mucosal microbiotas.

The nose can influence the conjunctival microbiota via the lacrimal duct, and so it can be presumed that bacteria or their components (including resistance genes) can be transported to the eyes, where they probably affect ocular immunology (the nose–eyes microbiota axis). However, it is not yet known whether bacterial resistance selected in the nose or eyes moves from one to the other.

Broad-spectrum antibiotic suppression of the microbiotas of different body sites (gut, vagina, and skin) increase the risk of colonization by resistant pathogens, supressing the susceptible ones [71]. Fluoroquinolones are among the most widely prescribed antibiotics in the world, which has been previously described, and their efficacy and epidemiological resistance in the pathogens have been studied in detail. However, a number of studies have shown that fluoroquinolones have dysbiotic effects and disturb microbial flora. Their negative qualitative and quantitative impact on the three main human microbiota has recently been discussed by de Lastours and Fantin [72], and de Lastours et al. [73]. Another study [74] has recently investigated the effects of different concentrations of ciprofloxacin and clindamycin on skin, saliva, nasal, and gut microbiotas. Their oral administration affected anaerobic intestinal microbiota, whereas they did not seem to have any effect on skin or oro-nasal microbiota. However, Munier et al. [75] studied nasal microbiota in 49 hospitalized and 62 community patients after treatment with fluoroquinolones and found that the acquisition of fluoroquinolone-resistant strains (particularly from extra-nasal sites) is frequent in the community and almost inevitable in hospitals.

To the best of our knowledge, there are no published data concerning the effect of topical drugs on human ocular microbiota. It can be expected that bactericidal drugs affect the microbiota more than bacteriostatic ones: however, this has yet to be proved. Chloramphenicol is bacteriostatic (or more precisely, bactericidal at certain concentrations and in the case of particular bacteria) [76] and should thus theoretically preserve the abundance and diversity of ocular microbial flora more effectively than topical bactericidal agents, such as the aminoglycosides or fluoroquinolones. However, further challenging studies are required to investigate the different effects of all of these agents on ocular or nasal microbiota and, especially, the ability of topical drops or ointments to select microbial resistance inside the microbiota.

## 6. Discussion

The high levels of antimicrobial resistance encountered throughout the world is due to the abuse, misuse, and overuse of antibiotics, and multi-drug resistance (MDR) to three or more antibiotic classes is also increasing in some countries and in the case of some infections. The studies mentioned above have shown that MRSA contributes to the pathology of various eye diseases, that hospitalization and previous antibiotic treatments accelerate the risk of MRSA colonization and infections [77], and that MRSA keratitis is a severe complication of post-refractive surgery. The ARMOR surveillance study demonstrated increasing resistance among staphylococcal isolates, with the MDR rates of MRSA and MR CoNS increasing from approximately 40% to 80%, especially in elderly patients.

Topical antibiotics are kinetically more effective in rapidly ensuring high concentrations at the site of infection than those systemically administered antibiotics. Although the topical administration of antibiotics may be better, prolonged treatments can also lead to drug resistance.

An old drug, such as chloramphenicol, is still a very good means of overcoming resistance: the rate of resistance to chloramphenicol is low and, in some cases, comparable with the still-low rate of resistances of *Enterococcus* to vancomycin.

It may also be true that high concentrations of topical antibiotics can eradicate strains that are resistant in-vitro and that the breakpoints of systemic antibiotics cannot be applied to their topical counterparts [78]. However, antibiotic resistance is becoming increasingly prevalent when treating eye infections, especially in the case of fluoroquinolone treatments. The inappropriately widespread use of the older fluoroquinolones (ofloxacin, ciprofloxacin, and levofloxacin) has led to the increased resistance of many “superbugs”, which can develop rapidly as a result of single- or multi-step mutations [79]. Multi-step mutations are more likely to occur after repeated bacterial exposure to low antibiotic levels or the use of intermittent or excessively long-term treatments [80].

Another important aspect of eye infections is the contribution of bacterial biofilms, and Behlau and Gilmore [47] have clearly described the mechanism underlying biofilm formation and provided a list of biofilm-related eye infections. Understanding how bacteria resist the activity of antibiotics when embedded in a biofilm would be a very useful step towards bacterial eradication, even though matrix-embedded cells are often simply inaccessible. There are few data concerning the activity of ocular antibiotics against biofilm-related infections but, despite the lack of any clear in-vivo evidence, it is known that chloramphenicol can interfere with bacterial adhesion before biofilm formation, and there is a good chance that it is also capable of penetrating the biofilm matrix. It could therefore represent a means of combating biofilm-related infections and improving patient outcomes, although further studies are required to demonstrate its effectiveness.

The genomic-based detection and identification of microbial species has significantly extended our knowledge of eye infections and has also shown the presence of surface microbiota. A deep sequencing study of conjunctival DNA revealed an average of 221 species of bacteria per analyzed subject: these could be classified into five phyla and 59 distinct genera, twelve of which were found in all subjects [81]. The existence of resident ocular surface microbiota suggests that they play a protective role in preventing the proliferation of pathogenic species, and that eye diseases may be related to alterations in microbiota homeostasis. The disruption of normal ocular microbiota may be a significant co-factor in the pathogenesis of ophthalmic diseases. There is therefore a need for antibiotics that are not only capable of combating the bacteria involved in contact lens-associated corneal infiltrative events, blepharitis, and post-operative infectious endophthalmitis, but also capable of gently treating the bacterial neighbors harboring ocular microbiota. It is likely that bacteriostatic antibiotics are more suitable for preserving good ocular bacteria, but this needs further investigation.

Clinicians should choose antibiotic formulations on the basis of the location and severity of the infection, the epidemiology of regional and local bacterial resistance, their ability to act against biofilms, and the patient’s co-morbidities. Topical chloramphenicol formulations are probably trying to satisfy almost all of these requirements (see Table 1), even if more in-depth studies are needed to fully clarify its clinical role in eye infections.

## Figures and Tables

**Table 1 microorganisms-07-00278-t001:** Microbiological characteristics of chloramphenicol in 2019′s eye infections.

Characteristics	Activity	References
Spectrum of Activity	S.aureus (MRSA) S.epidermidis (MRSE) CoNS Streptococcus Pseudomonas Corynebacteria	[39,40,41,42]
Antibiofilm activity	Biofilm penetration Biomass reduction Adhesion interference	[50,59,60,61,62,63,64]
Ocular Microbiota protection *	Bacteriostatic vs. Bactericidal microbiota disturbance	[18,67,68,70,72,73,74,75,76]

* Need to be evaluated.

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
