# Peer review of "Chloramphenicol Resurrected: A Journey from Antibiotic Resistance in Eye Infections to Biofilm and Ocular Microbiota"

_microorganisms, 2019, doi:10.3390/microorganisms7090278_

Round 1

Reviewer 1 Report

The papers covers an interesting subject. We are all aware of the fact, that due to drug resistance we are having troubles with antibiotics and treatment, but showing it in this particular filed, which is ocular microbiota and eye infection is a suprising, yet interesting concept. The data may be useful for both - clinicists and sicentists. The most valuable (in my opinion) part of the manuscript is the biofilm's role in eye infections - the data should be popularised also among patients using contact lenses for example. Understanding the role of microbiota is now not only crucial but also a widely commented subject, that is why I think the paper is worth publishing. There are some small spelling mistakes that have been overlooked and I would like to ask the Authors to be consistent in using italics for bacteria species. 

I recommend the paper to be printed.

Author Response

Reviewer 1

The papers covers an interesting subject. We are all aware of the fact, that due to drug resistance we are having troubles with antibiotics and treatment, but showing it in this particular filed, which is ocular microbiota and eye infection is a suprising, yet interesting concept. The data may be useful for both - clinicists and sicentists. The most valuable (in my opinion) part of the manuscript is the biofilm's role in eye infections - the data should be popularised also among patients using contact lenses for example. Understanding the role of microbiota is now not only crucial but also a widely commented subject, that is why I think the paper is worth publishing. There are some small spelling mistakes that have been overlooked and I would like to ask the Authors to be consistent in using italics for bacteria species.

I recommend the paper to be printed.

Answer

I’d like to thank the Reviewer for positive comments.

Mistakes are now corrected along the manuscript as well as bacteria nomenclature which is now in italic.

Reviewer 2 Report

The proposed manuscript presents a review regarding the use of ocular topical chloramphenicol against biofilm-related infections. Besides, the author also discusses and the impact of this old drug, which still exhibits low rates of resistance, on the ocular microbiota. The paper is organized thorough the presentation of the topic. I appreciate the effort involved in compiling the literature and writing this review. The manuscript highlights both relevant and exciting insights in its possible implications for the prevention/treatment of biofilm-related ocular infections.

Major comments:

English needs major revision. Sentences are too long in some parts of the paper. There are mistakes in 3rd person plural/singular, among others.

There are many mistakes in taxonomic nomenclature (some specified in minor comments).

The references in the text are totally inaccurate.

There is a complete lack of correspondence between citations and the references list (at least from the reference [50] and throughout the rest of the manuscript). Besides, the reference list must be corrected and adapted to the journal' style.

I think it is important to add, even if only in the introduction, a description of controversies related to the use of chloramphenicol. In particular on the resistance or decreased bacterial sensitivity to chloramphenicol, and the mechanisms linked to the potentially toxic effects in humans. Indeed, Chloramphenicol is often blamed for being a possible cause of aplastic anemia [Yunis AA, et al. Chloramphenicol toxicity: Pathogenetic mechanisms and the role of the p-NO2 in aplastic anemia. Clin Toxicol 1980;17:359–373]. Although, one of the best-documented cause of AA is with viral hepatitis. [Brown, K.E., et al. Hepatitis-associated aplastic anemia. New England Journal of Medicine 1997;336,1059–1064; Hepatitis associated aplastic anemia: a review. Virol J. 2011 Feb 28;8:87.]. The mechanism of action is believed to result from abnormal metabolism of this molecule. Reduction of the nitrobenzene ring to a nitroso group on chloramphenicol have the potential to cause DNA damages in the stem cell, and potentially cell death. Interestingly, other studies have hypothesized that gastrointestinal bacteria may metabolize chloramphenicol to toxic metabolites [Jimenez et al, Chloramphenicol-induced bone marrow injury: Possible role of bacterial metabolites of chloramphenicol. Blood 1987;70:1180–1185].

Specific comments are as follows:

Title:

Line 2: please change "BIOFILM" to "Biofilm".

Abstract:

Line 11: please change "unavailability" to "the unavailability"

Line 14-16: This sentence presents a parallelism problem due to the presence of the gerund "studying" followed by the to-infinitive "to optimize". Consider changing one of the verb forms to match the other one. Besides, change "use" to "the use".

Line 19: please change "peculiarity" to "peculiarities"

Line 25: please change "betteraddressed" to "better addressed"

Introduction:

Line 38-42: all bacterial species have to be italicized

Line 68: please change "thesystemic" to "the systemic"

Chapter 2:

Line 88: please change "micro-organisms" to "microorganisms"

Line 89: please change "and/o" to "and/or"

Line 90-95: "S.aureus, P.aeruginosa and S.epidermidis" have to be italicized with a space between the capital letter of the genus and the species name. Similar errors are present throughout the manuscript. Please correct.

Line 96-97: In this case, Staphylococcus epidermidis and Pseudomonas aeruginosa have to be italicized, and the genus indicated only with the capital letter since they have cited before. Similar errors are present throughout the manuscript. Please correct.

Chapter 3:

Line 116: please change the title "OF" to "of"

Line 120: change "programmes" to "programs" and line 122 "programme" to "program"

Line 122: please change "in vitro" to "in-vitro"

Line 126: please change "anti-microbial" to "antimicrobial"

Line 165: please change "fluoroquinolones has" to "fluoroquinolones have"

Line 141: please change "micro-organisms" to "microorganisms"

Chapter 4:

Line175: please change "intra-ocular" to "intraocular"

Line 196: revise the references at least from the N50 throughout the rest of the manuscript.

Line 197, 210, 224: incomplete hyphenation for the word "antimicrobial" in this part of the manuscript.

Line 230: "Chloramphenicol was more active against S. aureus and tobramycin more active", this sentence includes an incomplete comparison. Consider rewriting.

Chapter 5:

Line 248: consider changing the word order "an immunological ocular response" to "an ocular immunological response"

Line 267: please change "increase" to "increases" and "…pathogens by suppressing" to "…pathogens suppressing".

Line 270: please change "fluoroquinoloneshave" to "fluoroquinolones have"

Discussion

Line 295: please change "…is a severe complications" to "…is a severe complication"

Line 295: please change "post refractive" to "post-refractive"

Line 299-302: The sentence is too long, and t the word antibiotic/s is repeated 4 times. Consider changing.

Line 309: please change "…especially In the" to "…especially in the"

Be more critical and not so assertive regarding the use of chloramphenicol in the conclusions.

Author Response

REVIEWER 2

The proposed manuscript presents a review regarding the use of ocular topical chloramphenicol against biofilm-related infections. Besides, the author also discusses and the impact of this old drug, which still exhibits low rates of resistance, on the ocular microbiota. The paper is organized thorough the presentation of the topic. I appreciate the effort involved in compiling the literature and writing this review. The manuscript highlights both relevant and exciting insights in its possible implications for the prevention/treatment of biofilm-related ocular infections.

Answer by the Author:

Thank you so much for this appreciation.

 Major comments:

English needs major revision. Sentences are too long in some parts of the paper. There are mistakes in 3rd person plural/singular, among others.

Answer by the Author:

The manuscript has been re-edited and mistakes corrected.

There are many mistakes in taxonomic nomenclature (some specified in minor comments).

Answer by the Author:

Thank you. They are now corrected along the manuscript.

The references in the text are totally inaccurate.

There is a complete lack of correspondence between citations and the references list (at least from the reference [50] and throughout the rest of the manuscript). Besides, the reference list must be corrected and adapted to the journal' style.

Answer by the Author:

The references are now further checked and adapted to the journal format. Additional references are also added as indicated by the Reviewer

 I think it is important to add, even if only in the introduction, a description of controversies related to the use of chloramphenicol. In particular on the resistance or decreased bacterial sensitivity to chloramphenicol, and the mechanisms linked to the potentially toxic effects in humans. Indeed, Chloramphenicol is often blamed for being a possible cause of aplastic anemia [Yunis AA, et al. Chloramphenicol toxicity: Pathogenetic mechanisms and the role of the p-NO2 in aplastic anemia. Clin Toxicol 1980;17:359–373]. Although, one of the best-documented cause of AA is with viral hepatitis. [Brown, K.E., et al. Hepatitis-associated aplastic anemia. New England Journal of Medicine 1997;336,1059–1064; Hepatitis associated aplastic anemia: a review. Virol J. 2011 Feb 28;8:87.]. The mechanism of action is believed to result from abnormal metabolism of this molecule. Reduction of the nitrobenzene ring to a nitroso group on chloramphenicol have the potential to cause DNA damages in the stem cell, and potentially cell death. Interestingly, other studies have hypothesized that gastrointestinal bacteria may metabolize chloramphenicol to toxic metabolites [Jimenez et al, Chloramphenicol-induced bone marrow injury: Possible role of bacterial metabolites of chloramphenicol. Blood 1987;70:1180–1185].

Answer by the Author:

Thank you for these suggestions. Despite the topic is mainly focused on ocular infections and topical use of chloramphenicol, you may find these considerations in the Introduction session. The references indicated as well as those regarding the antibiotic resistance of chloramphenicol are now added.

Specific comments are as follows:

Title:

Line 2: please change "BIOFILM" to "Biofilm".

Done

 Abstract:

Line 11: please change "unavailability" to "the unavailability"

Done

Line 14-16: This sentence presents a parallelism problem due to the presence of the gerund "studying" followed by the to-infinitive "to optimize". Consider changing one of the verb forms to match the other one. Besides, change "use" to "the use".

Thank you. The entire manuscript is now revised in the English form as well as the related mistakes.

Line 19: please change "peculiarity" to "peculiarities"

Done

Line 25: please change "betteraddressed" to "better addressed"

Done

Introduction:

Line 38-42: all bacterial species have to be italicized

Done

Line 68: please change "thesystemic" to "the systemic"

Done

Chapter 2:

Line 88: please change "micro-organisms" to "microorganisms"

Line 89: please change "and/o" to "and/or"

Done

Line 90-95: "S.aureus, P.aeruginosa and S.epidermidis" have to be italicized with a space between the capital letter of the genus and the species name. Similar errors are present throughout the manuscript. Please correct.

Done

Line 96-97: In this case, Staphylococcus epidermidis and Pseudomonas aeruginosa have to be italicized, and the genus indicated only with the capital letter since they have cited before. Similar errors are present throughout the manuscript. Please correct.

Right. They are now corrected.

Chapter 3:

Line 116: please change the title "OF" to "of"

Line 120: change "programmes" to "programs" and line 122 "programme" to "program"

Line 122: please change "in vitro" to "in-vitro"

Line 126: please change "anti-microbial" to "antimicrobial"

Line 165: please change "fluoroquinolones has" to "fluoroquinolones have"

Line 141: please change "micro-organisms" to "microorganisms"

All these changes are now made.

Chapter 4:

Line175: please change "intra-ocular" to "intraocular"

Line 196: revise the references at least from the N50 throughout the rest of the manuscript.

Line 197, 210, 224: incomplete hyphenation for the word "antimicrobial" in this part of the manuscript.

Line 230: "Chloramphenicol was more active against S. aureus and tobramycin more active", this sentence includes an incomplete comparison. Consider rewriting.

All these changes are now performed.

Chapter 5:

Line 248: consider changing the word order "an immunological ocular response" to "an ocular immunological response"

Line 267: please change "increase" to "increases" and "…pathogens by suppressing" to "…pathogens suppressing".

Line 270: please change "fluoroquinoloneshave" to "fluoroquinolones have"

These changes are now considered in the text.

Discussion

Line 295: please change "…is a severe complications" to "…is a severe complication"

Line 295: please change "post refractive" to "post-refractive"

Line 299-302: The sentence is too long, and t the word antibiotic/s is repeated 4 times. Consider changing.

Line 309: please change "…especially In the" to "…especially in the"

All these changes are now made.

Be more critical and not so assertive regarding the use of chloramphenicol in the conclusions.

The sentence has been modified accordingly.

Round 2

Reviewer 2 Report

Accept the article in the present form

This manuscript is a resubmission of an earlier submission. The following is a list of the peer review reports and author responses from that submission.